# Acceptance of Social Networking Sites by Older People before and after COVID-19 Confinement: A Repeated Cross-Sectional Study in Chile, Using the Theory of Planned Behaviour (TPB)

**DOI:** 10.3390/ijerph192013355

**Published:** 2022-10-16

**Authors:** Patricio Ramírez-Correa, Elizabeth Grandón, Jorge Arenas-Gaitán, Javier Rondán-Cataluña, Muriel Ramírez-Santana

**Affiliations:** 1Engineering School, Universidad Católica del Norte, Coquimbo 1780000, Chile; 2Department of Information Systems, Universidad del Bío-Bío, Concepción 4081112, Chile; 3Department of Business Administration and Marketing, Universidad de Sevilla, 41018 Sevilla, Spain; 4Public Health Department, Faculty of Medicine, Universidad Católica del Norte, Coquimbo 1780000, Chile

**Keywords:** social networking sites, older people, theory of planned behaviour, cross-sectional study, Chile

## Abstract

This study aims to examine the capacity of the Theory of Planned Behaviour (TPB) to explain the intention to use social networking sites by older people in two time periods, before and after confinement due to the COVID-19 epidemic, as well as the evolution of effects (paths) over time of TPB’s determinants. Based on interviews from samples of 384 and 383 elderly Chilean adults collected before and after confinement, the evolution of the effects (paths) was analysed using the TPB model applying the PLS-SEM technique. The intention to use social networks and its association with three factors were evaluated: attitude toward the behaviour, subjective norms, and perceived control over the behaviour. The model explains the intention to use social networks by 27% before confinement, increasing its magnitude to 50% after confinement. After the period of confinement, their attitudes become more significant, their perceptions of control become less important, and social pressures remain permanent in predicting the behaviour. In conclusion, better access and greater use of social networks by older people during the lockdown period increased the predictive strength of the attitude towards these technologies.

## 1. Introduction

The use of information technologies has a positive impact on the well-being of the elderly. Among its health benefits, the improvement of cognitive performance and physical condition to carry out activities of daily living have been described, promoting active aging and social participation [1,2]. However, before the confinement of COVID-19, there was a big gap in Internet use by older people in Chile. In 2014, 66% did not use the Internet, a measure that remained the same until 2017 (54% of access in households in which only people over 64 live). This gap increases as income level decreases [3]. In general, and prior to the COVID-19 epidemic, barriers to the use of digital technologies had been identified in this group of people related to psychological and social issues, such as motivation, attitude, privacy, trust, and learning on the use of technology [1]. Regarding stimuli, particularly in Chile, it has been described that close to 40% of older people had hedonic motivations for using social networks, without differences by sex, before confinement [4].

The increased risk of severity and death from COVID-19 infection for older people translated into much stricter confinement for this population group in the first two years of the pandemic. As a result, the lockdown in Chile was established from the last week of March 2020 to the end of July 2021, with a variable modality, according to the epidemiological indicators of each region and city of the country. In this seclusion, the use of social network technologies increased in this group of people, allowing them to mitigate the effects of social isolation. Indeed, a study of the use of digital technologies in the Chilean elderly, carried out in late 2021 by a communications company, showed that 40% learned to use smartphones during the pandemic and that only 8% felt limited in their activities for not knowing how to use the Internet. Likewise, 91% of older people had a personal computer and 90% used it regularly. Moreover, 91% considered that the Internet is useful in their daily lives, 89% looked for movies or news, and 86% connected daily to social networks. The social networks most used by them were WhatsApp (95%), Facebook (82%), and YouTube (60%); 80% carried out procedures online, and of these, only 14% would like to carry them out in person again [5]. However, although the pandemic may have increased access and been a strong incentive for using these technologies, it is not clear whether, when these restrictions are lifted, the elderly will continue to use them with the same intensity or whether they will return to their previous state, that is, if the lockdown has influenced the process of adoption of online social networks by the elderly.

Specifically, differences in technology adoption have been explained mainly by two research streams. On the one hand, the Bass model tries to provide a long-term forecast of the growth of a new product on the market [6]. This model considers two effects: (1) The external influence, called innovation, whose origin is the intrinsic tendency of the individual to adopt, and the effect of an external source of mass communication; and (2) The internal factor, called imitation, which derives from personal contact with previous adopters. Alternatively, the Nolan model explains how technology goes through various stages over time [7]. This model states the notion of learning in each stage, establishing four phases: initiation, contagion, control, and integration. Nolan’s model indicates that people can adopt technology at different stages of growth.

Within this context, several authors have explored the change in attitude strength as a determinant of the intention to use information technology. A study by Karahanna et al. (1999) was one of the first to distinguish between pre-adoption and post-adoption for the effect of the attitude on the intention to use information technologies; their results report that in post-adoption, the effect of attitude is stronger than in pre-adoption [8].

Most of the studies focus on describing the benefits and barriers to using digital technology for older people. In the case of social networks, it is necessary to analyse the intention and use by this age group through theoretical models of technology acceptance based on the beliefs and attitudes of the subjects [4]. In addition, based on these theoretical models, follow-up studies are required to determine the explanatory capacity of these models to examine the pandemic’s consequences on older people’s perceptions.

The Theory of Planned Behaviour (TPB) assumes that an individual’s behaviour is influenced by the intention to perform the behaviour in question [9]. According to Ajzen (1991), “intentions capture the motivational factors that influence behaviour” (p. 181). Thus, the intention is determined by three factors: attitude toward behaviour (ATT), subjective norms (SN), and perceived behavioural control (PBC). Attitudes toward behaviour refer to the degree to which a person has a favourable or unfavourable attitude toward the behaviour in question. The subjective norm corresponds to the perceived social pressure to perform or not perform the behaviour. Finally, the perception of behavioural control reflects the perceptions that personal and situational factors may affect the ability to carry out the behaviour. Ajzen [9] pointed out that the more favourable the attitude and subjective norm regarding a behaviour, and the greater the perceived behavioural control, the stronger an individual’s intention must be to perform the behaviour under consideration. The theory also posits that perceptions of behavioural control directly determine behaviour when the person perceives that he or she has complete control to execute the behaviour.

Despite the breadth and variety of studies that have used TPB, there is a lack of research that validates this theory in longitudinal or pseudo-longitudinal studies. Some exceptions are Plotnikoff, Lubans, Trinh, & Craig (2012), Leung (2019), Roux, Gourlan, & Cousson-Gélie (2021), Thaker & Ganchoudhuri (2021), and Liu et al. (2022), who examined different behaviours and target populations [10,11,12,13,14]. Appendix A shows the path coefficients (B, betas) found in each relationship of the TPB at the different times considered and the determination coefficients R^2^ that explain the percentage of variance in the intention to execute the behaviour in question [10,11,12,13,14]. In general, the attitude, subjective norm, and perceptions of control predict the intention to execute the behaviour in question. However, people’s perceptions of these same constructs may change in intensity from one time to another. Similarly, the percentages of explained variance vary between times. These changes do not show a clear trend in the studies reviewed.

Following the technology adoption model, before the epidemic, older people in Chile who were initially using social networks were the group of “innovators”. Assuming that the need was generated during confinement, the elderly increased their use substantially (contagion stage), with the possibility of having modified the attitude of these people towards technology. At the same time, after confinement, older adults have greater knowledge and resources for using social networks, given family investments and social support programs. Therefore, the perceived behavioural control variable could lose its predictive power to explain the variance of the intention to use social networks.

In the context of technology adoption models, the objective of this study is to examine the capacity of the TPB to explain the intention to use social networks by older people in two periods of time, before and after COVID-19 confinement, as well as the evolution of the effects over time of its determinants. Based on the TPB, the hypothesis proposed by the researchers is:

H: There are significant differences in the strength of the determinants of the intention to use social networks by older people before and after confinement due to COVID-19, so the attitude increases its predictive strength and the subjective norms maintain its predictive strength, while perceived control decreases it. Figure 1 shows the research model based on TPB.

## 2. Materials and Methods

### 2.1. Design and Sample

We use a repeated cross-sectional design, which is also known as pseudo-longitudinal. This type of survey design and use of data requests the same information from an independent sample in each cycle [15]. Using a face-to-face survey and measurement scales validated in the literature, we obtained data from older people who use social networks based on the research model. The inclusion criteria were age of sixty or more and being a user of social networks. People with cognitive impairment were excluded.

Additionally, age, gender, education, work activity, socioeconomic group, and experience in Internet use were assessed. The data collection procedure was carried out in two moments, the first (T1) in October 2019 and the second (T2) in September 2021. The regions of Coquimbo and Biobío were considered, since they present a higher proportion of the population over 60 years of age compared to Chile. Due to the nature of the proposal and the analysis of all elements of the population, a stratified sampling was carried out with similar calculation bases for each moment. As a sampling frame, the results of the most up-to-date population census available in the country for the regions of Coquimbo and Biobío were used. To determine the proportion of Internet users by gender and age range, the results of the “IX Internet Access and Use Survey of the Undersecretary of Telecommunications of Chile” were used, whose information constitutes the most up-to-date study in Chile [3]. Users were those who reported that they had used the Internet for the past three months or less. After filtering this database and calculating the proportions, the total number of elderly people was obtained. The total sample size was calculated using the random sampling formula for a finite sample:$$n=\frac{Nz_{\alpha /2}^{2}s^{2}}{e^{2}\left(N-1\right)+z_{\alpha /2}^{2}s^{2}}$$
where “*n*” is the size of the global sample, “*N*” is the size of the population defined in the sampling frame, “*Z*” is the critical value of a standardized normal random variable for a confidence level of 1 − *α*, “*s*” is the variance of the answers, and “*e*” is the maximum error allowed for the estimators of interest. Because the indicators of interest are proportions (scales of measurement), for the calculation, the maximum variance of a Bernoulli process was adopted, which is equal to 0.25 (the variance of a Bernoulli process is equal to *p*(1 − *p*), where “*p*” corresponds to the probability of success; therefore, the maximum value of this variance is reached with *p* = 0.5). With this, the calculation of the size of the global sample for the year 2019 was reduced to 384 and, for 2021, to 383. As a second step, the global sample was distributed in age and sex strata by simple affixation. Table 1 shows the distributions of the stratified samples according to the affixation described.

### 2.2. Measurements

The measurement was based on a survey of 112 questions, divided into four sections, plus a section of socio-demographic data. Specifically, to measure attitude towards behaviour, we used two items based on Kwon et al. (2014) (for example, “I think using social networking sites is beneficial to me” [16]). To measure subjective norms, we used three items based on Sun et al. (2014) (for instance, “People who influence my behaviour think that I should use social networking sites” [17]). To measure perceived behavioural control, we used two items based on Venkatesh et al. (2012) (for example, “I have the necessary knowledge to use social networking sites” [18]). Finally, to measure intention to use social networks, we used three items based on Venkatesh et al. (2003) (for instance, “I intend to use social networking sites in the next six months” [19]). All scales were measured on a 5-point Likert-type scale, with 1 indicating strong disagreement and 5, strong agreement with the statement.

### 2.3. Data collection and Data Analysis

Interviewers carried out the data collection in places with an influx of older people who use social networks, such as older adult programs at universities, public offices, and medical centres, among others. The surveyors associated mainly with health professions were trained for the data collection process. Eighteen interviewers participated in the first evaluation and twenty-three in the second. The interview lasted approximately 30 min and was conducted face-to-face in T1 and T2. They were also made by video call or telephone call, depending on participant’s availability. The interviewers entered the responses in a digital platform designed for this purpose, which allows the information to be recorded on the institutional server. All participants approved their involvement through informed consent. The Ethics Committee of the Universidad Católica del Norte approved the study protocol (R14/2019 and R05/2021).

For data analysis, the variance-based structural equation modelling (PLS-SEM) technique was used [20]. This technique examined both the measurement model and the structural model. The analysis was performed for each model (T1 and T2). Following the procedure proposed by Roemer (2016) [21], to examine the existence of significant evolutions in the path coefficients between T1 and T2, the Partial Least Squares Multigroup Analysis (PLS-MGA) was used [22]. Specifically, the procedure proposed by Roemer (2016) [21] is a systematic view of how to use PLS-SEM in longitudinal studies. This procedure posits that if the main research objective is to examine the evolution of effects in a model over time and the database does not consist of panel data, but if the researchers have measured the same indicators at different times with different samples (repeated cross-sectional data), the following steps should be followed: set up the PLS-SEM models separately according to the different moments in time, perform multigroup analysis to test changes in path coefficients over time, and perform independent sample *t*-tests of the construct scores. On the other hand, PLS-MGA is an approach based on PLS-SEM for multigroup analysis proposed by Henseler (2012) [22]. This method is a non-parametric significance test for group-specific outcome differences based on bootstrapping results from the PLS-SEM model. For a given difference in path coefficients between two groups, a result is significant at the 5% probability of error level if the *p*-value of the test is less than 0.05 or greater than 0.95.

Before performing the PLS-MGA, measurement invariance was verified using the MICOM procedure [23]. MICOM, which stands for Measurement Invariance Evaluation, is a three-step method that allows analysis of the measurement invariance of models before performing multigroup analyses in PLS-SEM. The first step proposes to ensure, in the calculation of the models associated with each group, that all the parameters of the PLS-SEM analysis are the same. The second and third steps are based on permutations of the data. The successful evaluation of the second step allows performance of an analysis of the differences in path coefficients between two groups, and, in addition, the successful evaluation of the third step establishes a complete invariance among the groups evaluated. The SPSS software was used for descriptive analysis, and the SmartPLS 4 software (GmbH, Oststeinbek, Germany) was used for all other analyses [24]. In addition, *p* values > 0.001 were considered significant. The common-method variance was performed to ensure that no systematic bias influences this data. Harman’s single-factor criterion was used to investigate the common method bias. In T1, the results indicated that the principal component of a fixed factor explains 38.18%, less than 50%, of the variance. In T2, the results showed that the principal component of a fixed factor explains 47.89%, less than 50%, of the variance. Additionally, the highest correlation among the constructs in T1 and T2 is 0.43 and 0.67, respectively, which is less than the cut-off value of 0.9 suggested by Baggozzi et al. (1991). Therefore, there is no common-method variance in the data [25].

## 3. Results

### 3.1. Characteristics of the Participants

As a result of the data collection approach, response rates were high. In T1 the response rate was 86.68%, and in T2 90.33%. In T1, the sample comprised 384 individuals, 56% women, with an average age of 68.71 (SD 6.54). At this time, 67.45% of those surveyed lived in the Biobío region, and the rest in the Coquimbo region; 63.80% lived with a partner, 63.54% were retired, and 57.03% identified themselves as belonging to the middle-income class. In T2, the sample included 383 individuals, 55.6% women, with an average age of 68.69 (SD 7.75). At this time, 67.62% of those surveyed lived in the Biobío region and the rest in the Coquimbo region; 62.92% lived with a partner, 60.84% were retired, and 66.06% recognized themselves as belonging to the middle-income class.

### 3.2. Structural Models

The calculations of the measurement model were adequate for both periods (T1 and T2). The results of PLS-SEM are shown in Table 2 and Figure 2a,b. Based on the procedure proposed by Kock and Hadaya (2018) [26], at T1 and T2, the sample sizes reached the minimum number to provide the analysis with a statistical power of 80%. The goodness-of-fit of the models was evaluated through the SRMR index, with values for T1 and T2 of 0.057 and 0.055, respectively. These values meet the accepted criteria for this index (SRMR < 0.08). The complete fit measures for T1 were d ULS = 0.181, d G = 0.177, Chi-Square = 441.831, and NFI = 0.830. The complete fit measures for T2 were d ULS = 0.164, d G = 0.186, Chi-Square = 481.950, and NFI = 0.835.

In T1, the coefficients have a weak magnitude of explained variance, but in T2 they increase to a moderate magnitude, going from 27% before the pandemic quarantines to over 50% after it. The analysis shows positive path coefficients in most relationships, except between perceived behavioural control and intention to use social networks after confinement. After the pandemic quarantines, the perception of behavioural control is insignificant in explaining the variance of older people’s intention to use social networks. On the other hand, among the antecedents of the intention to use social networks, attitude is the primary determinant before and after confinement; it is followed, before confinement, by the perception of behavioural control, and after confinement, by the subjective norms.

### 3.3. Multigroup Analysis (MGA)

Before performing the PLS-MGA, the MICOM procedure was run to determine the measurement invariance. The MICOM procedure automatically set the configural invariance (Step 1) in the Smart PLS 4 software (GmbH, Oststeinbek, Germany). Table 3 shows the composite invariance correlation and confidence interval of the following steps. The correlation values for the composite invariance are equal to 1. The lowest value of 0.999 and the correlation values are more significant than the 5% quintile values with insignificant *p*-values (Step 2), allowing us to continue with the next step of MICOM. In Step 3, equality of means and equality of variances are evaluated. For only two constructs, the values of mean differences and variance differences were found between the confidence interval of 2.5% and 97.5% in both conditions. This analysis indicated that there was partial measurement invariance; therefore, multigroup analysis was possible.

As can be seen in Table 4, two paths have significant evolution in the path coefficients between T1 and T2.

The results confirm the hypothesis established in the study. There are significant differences in the strength of the determinants of older people’s intention to use social networks before and after COVID-19 confinement. At the same time, the effect of attitude increases significantly between before and after confinement. On the other hand, the effect of perceived behavioural control decreases significantly between before and after confinement, which means that the variable perceived behavioural control no longer predicts the intention to use social networks in this group of older people after confinement (R^2^ = 0.071 in Figure 2). Additionally, subjective norms maintain their predictive strength for the behaviour studied.

## 4. Discussion

The results obtained have allowed us to achieve the stated objective. Next, we will delve into the discussion of our results.

This study contributes to knowledge, given that longitudinal or semi-longitudinal studies on the use of information technologies using the TPB model found in the literature are scarce. Moreover, this research represents the first repeated cross-sectional study of the TPB to explain the use of social networks in a sample of older people in Chile over two years. Additionally, it evaluates the change experienced in older people’s behaviour regarding using social networks due to lockdowns. Consistent with the TPB theory and with previous studies [12,27], at T1 before confinement, attitudes, subjective norms, and perceptions of control were directly related to the intention to use social networks by older people. However, almost two years after experiencing the lockdown, the model was partially validated, as perceived behavioural control was found not predictive of intention to use social networks. Our results show that, at T2, attitude increases its predictive power. At the same time, perceived behavioural control not only decreases but is no longer able to predict older people’s use of social networks. For their part, the subjective norms do not undergo significant variations at times T1 and T2. The same effect was observed in three of the four longitudinal studies that evaluated the use of technologies under the TPB model [11,12,13].

The results are in line with what Yu et al. (2005) reported about adopting online systems. They observed a difference between inexperienced and experienced users in terms of the effect of attitude to explain the intention of use [28]. Furthermore, their results indicate that for experienced users, the effect of attitude is stronger for predicting behaviour.

Our results are also in line with the longitudinal study by Liu et al. (2022) who, using the TPB, concluded that perceived behavioural control does not affect the intention to use tablets after 24 weeks (T2) in Chinese patients with chronic diseases. They also showed that the strength of attitude to explain the intention to use tablets increases with the time of use of this technology [14].

Various factors underlie possible explanations of the findings. The isolation caused by confinement forced older Chileans to communicate through social networks and to use the Internet to carry out procedures, make purchases, and stay informed, among other motivations. Older people have gained access to and experience in social networks during confinement, which has caused them to feel more confident and that they have the knowledge, skills, and, in general, the resources to use them [7]. Due to the above, we found a modification in the strength of the variables that determine the intention to use social networks. Investigations in other countries coincide in reporting that the epidemic forced the elderly population to use more information technologies, including those related to health care services and applications for physical training and messaging [29,30]. Other researchers agree that older adults embraced digital technology and had sufficient knowledge and support, but felt apprehensive and faced financial barriers to technology adoption [31].

In addition to socioeconomic barriers, other variables that influenced the use of these technologies in older people during confinement were access to the Internet and previous use of video calls [29], level of education, digital literacy, and attitudes toward oneself aging [30,32,33]. According to international research, the benefits obtained were related to improvements in mental health, decreased loneliness, strengthening of physical capacity, mood, and memory enhancements [34,35]. A study of adherence to digital health programs for adults showed that those over 65 had better adherence to the digital platform than younger adults [36]. Older people’s greater confidence in the use of social networks favours them in terms of increasing their self-esteem, reducing isolation, and improving their mental health [1,2]. However, it has some disadvantages, among them increasing the vulnerability to being the target of scams or insecure situations. It remains to be seen if this group of people will continue to increase the use of technologies and social networks in the future, being of interest to continue investigating the subject, given that we verify that changes occur over time and with special situations such as the one experienced with the epidemic of COVID-19.

Although the change in the influence of the subjective norms was not significant, there is an increase in their impact on the intention to use social networks, which is probably explained by the importance of maintaining social and family life virtually.

The study has some limitations. The first is related to the fact that, when exploring the intention to use social networks, all the participants are users of this technology. Therefore, it is not possible to explore factors that could explain the non-use of this type of technology or that could have changed the intention from not using it to using it. A second limitation of the study lies in the low power of the model, exemplified in a β of 0.14 in the first measurement and probably explained by a low sample number. According to Sarstedt, performing PLS analysis requires large sample sizes to provide sufficient statistical power [37]. However, the β coefficients are higher in the second measurement, making the model more powerful. On the other hand, the total number of respondents is consistent with other studies in this same age group, and its scope restricts the generalization of the findings. Finally, another limitation is that the first study (T1) was not carried out bearing in mind that we would experience a pandemic situation, so the comparison of the model is limited to the constructs and questions asked in the first period. For example, it would have been interesting to consider factors such as geographical proximity to the family and social isolation, among others.

Finally, the results indicate that the determinants in the use of social networks by older people have changed after the experience of confinement. In this way, the policies developed in this area may consider these changes. In addition, the use of technologies by older people is a behaviour that is expanding in Chile and Latin America. Thus, under the technology adoption model, it is expected that there will be variation in the factors that predict said behaviour by this population group. This fact shows the need for policy decisions and strategies focused on older adults that consider the most recent information to ensure adequate implementation.

## 5. Conclusions

As older people begin to use social media to connect with family and loved ones during the 16 months of the lockdown period (from March 2020 to July 2021), their attitudes become more significant for predicting the behaviour of using social networks, their perceptions of control become less important, while the social pressures they may feel remain permanent over time. The greater importance of attitude and less importance of perceived control may reflect that older people have a better willingness to use social networks, learn how to use them, and feel more confident when using them.

The research reveals that the models which explain the intention to use social networks in older people can vary over time, especially with extreme situations that affect the strength of the determinants of this behaviour in particular. This must be considered to carry out timely evaluations when making decisions that affect users.

## Figures and Tables

**Figure 1 ijerph-19-13355-f001:**
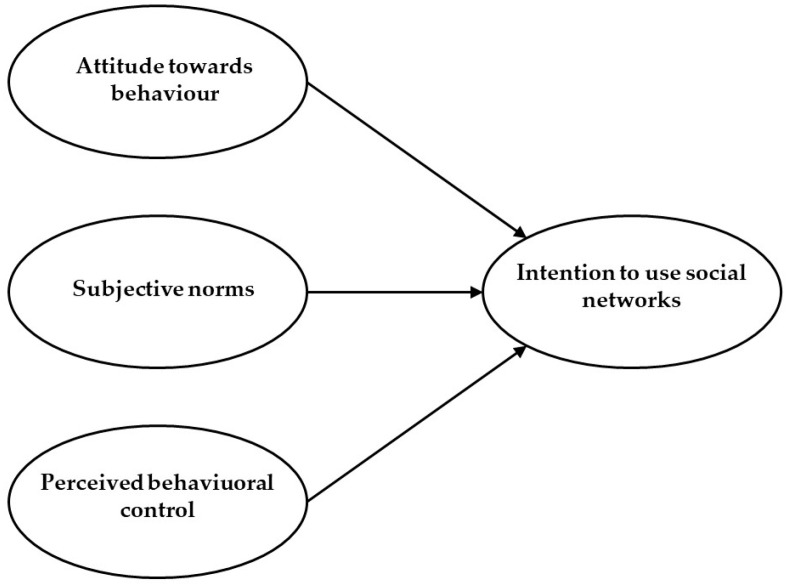
Research model, based on Theory of Plan Behaviour (TPB).

**Figure 2 ijerph-19-13355-f002:**
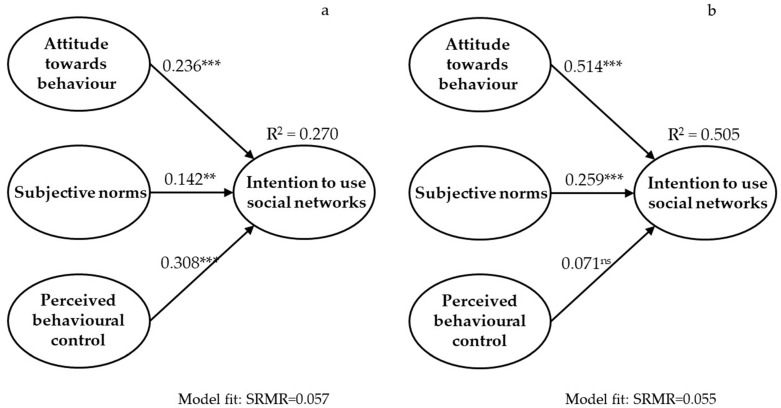
Changes of the Theory of Planned Behaviour models for elderly adults before (**a**) and after confinement (**b**). R^2^: coefficient of determination. ** *p*-value < 0.01, *** *p*-value < 0.001, ns: not significant.

**Table 1 ijerph-19-13355-t001:** Summary of age and sex distribution of the sample of participants, before and after confinement in elderly adults from Chile.

Age	T1: Before Confinement			T2: After Confinement		
	Women	Men	*n* (%)	Women	Men	*n* (%)
60–69	128	104	232 (60.4)	121	107	228 (59.5)
70–79	72	53	125 (32.6)	66	50	116 (30.4)
80+	15	12	27 (7.0)	26	13	39 (10.2)
Total (%)	215 (56.0)	169 (44.0)	384 (100.0)	213 (55.6)	170 (44.4)	383 (100.0)

**Table 2 ijerph-19-13355-t002:** PLS-SEM analysis of the factors in the TPB model before and after confinement in elderly adults from Chile.

Path		T1: Before Confinement				T2: After Confinement		
	ß	*p*-Value	Confidence Interval Corrected for ß Bias		ß	*p*-Value	Confidence Interval Corrected for ß Bias	
			2.5%	97.5%			2.5%	97.5%
Attitude towards behaviour ⇒ Intention to use social networks	0.236 ***	0.000	0.109	0.360	0.514 ***	0.000	0.419	0.601
Subjective Norms ⇒ Intention to use social networks	0.142 **	0.007	0.042	0.246	0.259 ***	0.000	0.174	0.345
Perceived behavioural control ⇒ Intention to use social networks	0.308 ***	0.000	0.185	0.425	0.071 ^ns^	0.097	−0.019	0.151

Bias-corrected confidence interval based on a 10,000-sample bootstrap procedure. ^ns^: not significant, ** *p*-value < 0.01, *** *p*-value < 0.001.

**Table 3 ijerph-19-13355-t003:** Results of the MICOM procedure in elderly adults from Chile.

Composite (Step 2)	Correlation Value		5.0% Quintile		Permutation (*p*-Values)			Composite Invariance		
Attitude towards behaviour	0.999		0.995		0.392			Yes		
Intention to use social networks	1.000		1.000		0.421			Yes		
Perceived behavioural control	1.000		0.989		0.951			Yes		
Subjective norm	1.000		0.998		0.628			Yes		
Measurement Invariance (Step 3)	Mean diffs.	2.5%	97.5%	*p*	Variance diffs.	2.5%	97.5%	*p*	EM	MI
Attitude towards behaviour	−0.014	−0.145	0.137	0.845	0.123	−0.298	0.298	0.461	Yes	Full
Intention to use social networks	0.216	−0.135	0.158	0.006	−0.049	−0.358	0.329	0.776	No	Partial
Perceived behavioural control	0.381	−0.145	0.132	0.000	−0.120	−0.266	0.252	0.360	No	Partial
Subjective norm	0.070	−0.145	0.145	0.353	0.012	−0.231	0.225	0.900	Yes	Full

Note: EM = equal means, MI = measurement invariance.

**Table 4 ijerph-19-13355-t004:** PLS multigroup analysis of the factors in the TPB model before and after confinement in elderly adults from Chile.

Path	T1 vs. T2	
	ß1–ß2	*p*-Value
Attitude towards behaviour > Intention to use social networks	−0.278 **	0.001
Subjective Norm > Intention to use social networks	−0.118 ^ns^	0.086
Perceived behavioural control > Intention to use social networks	0.237 **	0.003

ns: not significant, ** *p*-value < 0.01.

## Data Availability

Not applicable.

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
