# Peer review of "Acceptance of Social Networking Sites by Older People before and after COVID-19 Confinement: A Repeated Cross-Sectional Study in Chile, Using the Theory of Planned Behaviour (TPB)"

_ijerph, 2022, doi:10.3390/ijerph192013355_

Round 1

Reviewer 1 Report

Thanks for inviting me to review this paper. This study examined the changes in the effects of TPB variables on the intention to use social networks during the pandemic. Study background is well-written and clearly explains the theoretical basis of this study. However, I do not find any description of the sample inclusion criteria, study measurements, and demographic characteristics of the sample. This information is very important to evaluate the scientificity of this study. I hope the authors could add this information if any. My specific comments are as follows for the authors’ consideration:

  1. Title: Given the TPB was used to explore older adults’ acceptance of social networking; I think it can be added to the title.
  2. Abstract: There were no descriptions of measurements and study results in the abstract.
  3. p.1, Line 18: “In conclusion, as major people begin to use social network sites to connect with their families and people…”. Do you have any study results supporting this statement? or do you reach this conclusion based on previous literature?
  4. p.1, Line 34: please could the authors provide income ranges for quintiles 1 and 5?
  5. p.3, Line 122: The hypothesis is not clearly presented. Could the authors describe the direction of the change in the hypothesis? i.e., whether the coefficient of a variable in the model would increase or decrease from T1 to T2.
  6. Introduction: Could the authors provide background information about the policy responses to the pandemic in Chile? Any policy of lockdowns or restrictions in 2019 and 2021?
  7. Materials and methods: Did the authors follow any reporting guidelines when preparing this manuscript? I do not find descriptions of the inclusion and exclusion criteria of study samples and the measurements used. Please add if any.
  8. p.4, Line 150: May I know did the authors calculate the required minimum sample size for testing the model?
  9. Data analysis: Did the authors have any indicators for evaluating the goodness-of-fit of the model?
  10. Results: Please describe the demographic characteristics of the sample.
  11. Results: Please also provide the response rate of two cross-sectional surveys.
  12. Figure 2: please report the fitness of the models.
  13. Table 3: I think this table is not necessary. The R2 is shown in Figure 2.
  14. p.7, Line 234: May I know why the authors stated that the significant differences were caused due to COVID-19? Did the study results support this statement? Besides the pandemic, multiple factors may also affect one’s intention to use social networks.
  15. p.8, Line 298: May I know how long is the lockdown period in Chile?

Author Response

Dear reviewer: Please see the attachment where we provided the response point by point.

Reviewer 2 Report

This paper has the potential to make practical contributions, but some places need further elaboration. Below, I would like to suggest a few points that might be helpful for future revision.

1.      Line 30: It is not clear why the authors mention the “SARS-CoV-2 virus” at this point while the title mentions “COVID-19”. Please clarify. Consistent use of the term is recommended. Same in lines 15, 40, 72, and so forth.

2.      Line 37: What does it mean to have “hedonic motivations for the use of social networks”? In the previous sentence, the authors described barriers to the use of digital technologies by elderly. Without clear transition between the sentences, it is quite confusing why the authors suddenly describe motivations to use social networks.

3.      Line 81: The description of perceived behavioral control seems to be the opposite of the concept. The term does not refer to the “the perceptions that there are personal and situational impediments to carrying out the behavior”. Please revise.

4.      Hypothesis: The authors have not provided any logics underlying the hypothesis. Why do the authors predict that the strength of the determinants of the intention to use social networks would differ before and after COVID-19 confinement? Do the authors expect the strength to increase or decrease when compared between before and after the pandemic? Without logical reasons for the change and any clear directions of change, the current hypothesis does not seem to be appropriate.

5.      Measurement: Please explain the scales used to measure the variables. Please cite the scales, show examples of measurement items, the number of measurement items for each variable, and so forth.

6.      Data collection: How long did each interview last, on average? Was it conducted over phone or in-person? How many interviewers? Please explain.

7.      Line 230: What does “the effect of perception of control decreases” mean? Does it mean that the respondents have perceived less control after the pandemic? A clearer description is needed.

8.      Line 248: What was the reason in Yu et al. (2018) and Liu et al. (2022)’s studies for the insignificant effect of perceptions of behavioral control on intention to use tablets?

9.      Line 254: What does it mean for the “perceptions of control” to decrease significantly? Does it mean the respondents perceive their ability to control less than before? Please explain more clearly.

10.  Limitations: Because the study data relies on a single source, it is recommended to perform Harman’s single factor test to analyze any potential common method bias.

11.  Please specify the theoretical and practical contributions of this study.

Author Response

Dear reviewer: please see the attachment where we provided the response point by point. Thanks.

Reviewer 3 Report

Dear Authors,

I read the article with great interest. The issue you have taken up is very topical. Older people are usually seen as a social category at risk of digital exclusion. On the other hand, the global changes brought about by the COVID-19 pandemic appeared to have a significant impact on changing the very attitudes of older people towards new technologies. In my opinion, the article is not well developed. However, your paper has a very high potential, so it is worth the effort to make some modifications. My main concern is the structure of the content breakdown:

1. The paper has an overly elaborate introduction, while it lacks a theoretical section. The introduction should be shorter, giving only an overview of what the study will be about. At the end, the structure of the article should be described (e.g. the article is divided into chapter 1.... etc. or the paper is structured as follows: Section 1 provides....). The introduction should also not present the research methodology (description of hypotheses, methods should be moved to the methodology section).

2. In the article you should add a theoretical section and describe the theory, previous research results that will be referred to in the discussion (I know that you refer to the theory in the introduction, but this structure should be changed. In the introduction we only introduce the general outline of the issue).

3. The paper is missing a section on limitations of the research and implications for future research.

4. The discussion should refer more to the cited theory, previous research, once again refer to the authors cited in the theory section.

5) Conclusions - it seems to me that this part is too short and general. Surely, this part could be slightly expanded ( by at least 10 sentences).

I hope you find my comments helpful. 

Author Response

Dear reviewer, please see the attachment where we response poin by point your comments and suggestions. Many thanks.

Round 2

Reviewer 1 Report

Thanks to the authors’ effort put into the revised manuscript. Most of the comments have been appropriately addressed. I have some minor points for the authors’ consideration.

1. p.4, Line 172: Still, I do not find the inclusion criteria of the study sample. Are there any criteria to be included except age? Such as cognitive status, physical function, mental health problems, and having access to the internet.

2. p.5, Line 223: I do not find the statistical software/tools used for descriptive analysis and SEM. And please report the significance threshold, i.e., p-value, used in the methods.

3. p.7, Line 288: I would appreciate the SRMR was added. However, it was suggested that multiple indices should be reported in the assessment of model fit, such as RMSEA, the Chi-square test, CFI, etc. Please refer to Hooper, D., Coughlan, J., Mullen, M.: Structural Equation Modelling: Guidelines for Determining Model Fit. Electronic Journal of Business Research Methods, 6(1), 53-60. Also, please report these indices in data analysis methods.

Author Response

Reviewer 1

  1. 4, Line 172: Still, I do not find the inclusion criteria of the study sample. Are there any criteria to be included except age? Such as cognitive status, physical function, mental health problems, and having access to the internet.

R: We thank you the comment. We included a better description of the inclusion and exclusion criteria, as follows: The inclusion criteria were the age of sixty or more and being a user of social networks. People with cognitive impairment were excluded.

  1. p.5, Line 223: I do not find the statistical software/tools used for descriptive analysis and SEM. And please report the significance threshold, i.e., p-value, used in the methods.

R: Thanks for your comment. Although the sentence " SmartPLS software was used for all analyses [24]." was in line 217. We have enhanced this statement to add all the software used as follows:

“The SPSS software was used for descriptive analysis and the SmartPLS software was used for all other analyses. In addition, p- values>0.001 were considered significant.

As well, we added p-values in Table 2.

  1. p.7, Line 288: I would appreciate the SRMR was added. However, it was suggested that multiple indices should be reported in the assessment of model fit, such as RMSEA, the Chi-square test, CFI, etc. Please refer to Hooper, D., Coughlan, J., Mullen, M.: Structural Equation Modelling: Guidelines for Determining Model Fit. Electronic Journal of Business Research Methods, 6(1), 53-60.Also, please report these indices in data analysis methods.

R: Thanks for your comment. We include the report of all the indices that can be calculated in PLS-SEM. Now it says:

The goodness-of-fit of the models was evaluated through the SRMR index, with values for T1 and T2 of 0.057 and 0.055, respectively. These values meet the accepted criteria for this index (SRMR < 0.08).  The complete fit measures for T1 were d_ULS = 0.181, d_G = 0.177, Chi-Square = 441.831, and NFI =0.830. The complete fit measures for T2 were d_ULS = 0.164, d_G = 0.186, Chi-Square = 481.950, and NFI =0.835.   

Reviewer 2 Report

The authors have done an impressive job revising the manuscript. I have some additional comments for further revision.   1. The hypothesis still does not mention how 'subjective norms' may increase or decrease the intention to use social networks. Please revise. 2. The discussion section still needs to address how this research fills the gap in the literature. The discussion section only elaborates on details of the research findings but does not explicitly mention how these findings add to the literature.  3. The discussion section also needs to mention the study's practical implications. What policy implications can be suggested based on the research findings?

Author Response

Reviewer 2

The authors have done an impressive job revising the manuscript. I have some additional comments for further revision.  

  1. The hypothesis still does not mention how 'subjective norms' may increase or decrease the intention to use social networks. Please revise.

R: Thank you for the comment. We have modified the hypotesis: H: There are significant differences in the strength of the determinants of the intention to use social networks by older people before and after confinement due to COVID-19, so the attitude increases its predictive strength, the subjective norms maintain its predictive strength, and perceived control decreases it."

Results section and discussion were modified accordingly.

  1. The discussion section still needs to address how this research fills the gap in the literature. The discussion section only elaborates on details of the research findings but does not explicitly mention how these findings add to the literature.

R: Thank you for the remark. We commented on the contribution of our research in the second paragraph of the discussion:

This research represents the first repeated cross-sectional study of the TPB for the ex-planation of the use of social networks in a sample of older people in Chile, in a period of two years. Additionally, it evaluates the change experienced in the behaviour of older people regarding the use of social networks due to lockdowns.

We incorporated a new phrase at the biggening of the discussion, in relation to the contributions:

This study contributes to knowledge, given that longitudinal or semi-longitudinal studies on the use of information technologies using the TPB model found in the literature are scarce.

  1. The discussion section also needs to mention the study's practical implications. What policy implications can be suggested based on the research findings?

R: We thank you statement and added a paragraph in the discussion, as follows:

Finally, the results indicate that the determinants in the use of social networks by older people have changed after the experience of confinement. In this way, the policies that are developed in this area may consider these changes. In addition, the use of technologies by older people is a behaviour that is expanding in Chile and Latin America. Thus, under the technology adoption model, it is expected that there will be variation in the factors that predict said behaviour by this population group. This fact shows the need for policy decisions and strategies focused on older adults that consider the most recent information, to ensure adequate implementation.

Reviewer 3 Report

Dear Authors,

thank you for adding my suggestions. Now, the paper is ready to be published. I wish you many successes.

My kind regards, Reviewer

Author Response

We appreciate the good wishes and the fact that our answers have been satisfactory for you